# Essential Oil Blend Could Decrease Diarrhea Prevalence by Improving Antioxidative Capability for Weaned Pigs

**DOI:** 10.3390/ani9100847

**Published:** 2019-10-21

**Authors:** Qiyu Tian, Xiangshu Piao

**Affiliations:** State Key Laboratory of Animal Nutrition, College of Animal Science and Technology, China Agricultural University, Beijing 100193, China; qiyu.tian@wsu.edu

**Keywords:** antibiotics, antioxidative capacity, essential oil, diarrhea prevalence, weaned pigs

## Abstract

**Simple Summary:**

Antibiotics have been applied as growth promoters in swine production for many years. Due to increased concern about drug resistance, there is an urgent need to find alternatives to antibiotics for animal production. Present research indicates that essential oils have a beneficial influence on animal nutrition and production due to the antimicrobial and antioxidant properties. Dietary essential oil supplementation could be an alternative to antibiotics for improving swine production and decreasing diarrhea prevalence during the weaning period.

**Abstract:**

Finding an alternative to in-feed antibiotics is important because of increasing contemporary concern regarding drug residues and the development of drug-resistant bacteria. The purpose of this study was to test the hypothesis that essential oils added to the feed would decrease diarrhea prevalence in post-weaned pigs. Ninety weaned piglets (initial body weight (BW): 8.1 ± 1.4 kg) were randomly assigned to one of three dietary diets: (1) a control diet (CON, the basal diet without antibiotics), (2) an antibiotic diet (AB, CON supplemented with colistin sulfate, 20 mg/kg and bacitracin zinc, 40 mg/kg), or (3) an essential oil diet (EO, CON supplemented with an essential oil blend 100 mg/kg) in a completely randomized block design for a 28-day period. The results revealed that AB and EO improved the average daily gain of the piglets from day (d) 15 to 28 (*p* < 0.05). The diarrhea prevalence in piglets fed AB and EO was lower than that of piglets fed CON (*p* < 0.05). There was no significant difference in the growth performance or diarrhea prevalence between the AB and EO treatments. Nutrient digestibility was measured at d 28. Compared with CON, EO increased the apparent total tract digestibility of gross energy and crude protein (*p* < 0.05). Villus height in the duodenum and the ratio of villus height to crypt depth in the jejunum for piglets fed AB and EO was greater than those for piglets fed CON (*p* < 0.05). The essential oil blend improved the superoxide dismutase (SOD) and catalase (CAT) activities and total antioxidant capacity (T-AOC), but decreased the 8-hydroxy deoxyguanosine content in serum on d 14 (*p* < 0.05). Decreased malondialdehyde (MDA) and protein carbonyl content were observed on d 28 in comparison with CON (*p* < 0.05). The mucosa in the jejunum of pigs fed EO had greater T-AOC, SOD levels, and glutathione peroxidase (GSH-Px) activities than that of pigs fed CON (*p* < 0.05). Pigs fed EO and AB had greater GSH-Px activity in the liver tissue than pigs fed CON (*p* < 0.05). Not only did jejunal and ileal mucosa have EO upregulated SOD1 mRNA expression (*p* < 0.05), this was also the case in liver tissue. GPx1 expression in the ileal mucosa and GPx4 expression in the liver tissue were higher for pigs fed EO when compared to those fed CON (*p* < 0.05). Collectively, a dietary essential oil blend supplementation, which has natural antimicrobial properties, could enhance growth performance and decrease diarrhea prevalence in weaned pigs through increases in antioxidative capacity.

## 1. Introduction

Diarrhea is a major challenge for weaned pigs, which results in intestinal disturbances, performance depression, and immune dysfunction [1,2,3]. In-feed antibiotic growth promoters (AGP) have widely been used in young piglet diets to alleviate the side impacts of the high diarrhea prevalence caused by weaning stresses [4,5]. Currently, customers have been paying increasingly more attention to antibiotic residues in animal products, which may cause antibiotic resistance in zoonotic bacterial pathogens. The utilization of antibiotics in feeds as a growth promoter has been banned in the European Union since 2006. In July 2016, the use of colistin sulfate was prohibited for animal production in China. Essential oils with antimicrobial, antifungal, and antiviral properties have recently been considered as promising alternatives to antibiotics in feeds [6,7].

Essential oils are aromatic, volatile, and oily liquids synthesized by plants to deter herbivorous insects and animals [3,4]. These essential oils are generally considered natural, less toxic, and free from toxic residues when compared to antibiotics [8]. Various researchers have supported essential oils as alternatives to in-feed antibiotics for improving the performance and health of animals [9,10]. Our previous studies also indicated that carvacrol and thymol, as the two predominant components of essential oils, could enhance pig performance, nutrient digestibility, and the intestinal ecosystem [4,11,12]. However, the results obtained from previous studies are not always consistent [3], and less attention has been paid to the antioxidative ability of essential oils in alleviating the diarrhea prevalence of weaned piglets.

Therefore, the objective of this study was to test if decreased diarrhea prevalence was associated with the increased antioxidative capacity and growth performance of weaned pigs when administered a blend of essential oils. Growth performance, diarrhea prevalence, and fecal nutrient digestibility, as well as the activity and gene expression of antioxidant enzymes, were evaluated to assess the effects of dietary treatments. This study will potentially help decrease the use of antibiotics for weaned pigs and provide an alternative solution for sustaining the efficiency of current livestock production.

## 2. Materials and Methods

Our experimental procedures in the present study were approved by the Ethical Committee of China Agricultural University, and were in agreement with the China Agricultural University Laboratory Animals Welfare and Animal Experimental Ethical Inspection (Beijing, China; No. AW09089102-1).

### 2.1. Essential Oil Product

The commercial essential oil blend was provided by DuPont Co., Ltd. The product contained 4.5% cinnamaldehyde, 13.5% thymol, and 82% feed grade carrier in this study. Bioactive components were synthetic and encapsulated in a maltodextrin matrix.

### 2.2. Animal Experimental Design and Treatments

A total of 90 weaned piglets (Duroc × Landrace × Yorkshire) with similar body weight (8.1 ± 1.4 kg) were randomly assigned to one of three treatments with 6 pens (5 pigs/pen) per treatment according to sex and weight in a randomized complete block design. The dietary treatments included: (1) control diet (CON, the basal diet without antibiotics), (2) antibiotic diet (AB, CON supplemented with colistin sulfate, 20 mg/kg and bacitracin zinc, 40 mg/kg) and (3) essential oil diet (EO, CON supplemented with essential oil blend, 100 mg/kg). The diets were formulated to meet or exceed nutrient demands according to the National Research Council (NRC) (2012) standards [13] (Table 1). All diets were fed in mash form and contained 0.25% chromic oxide as an indigestible marker.

The experiment was designed with two periods (period I: post-weaning days 1 to 14; period II: post-weaning days 15 to 28). The weaned pigs were housed in pens with plastic slotted floors. Each pen contained a nipple drinker and a four-hole self-feeder to provide ad libitum access to water and feed. The temperature (24–26 °C) and humidity (60–70%) in the pig house was automatically controlled. The pens and barn were cleaned by broom every day to maintain a hygienic environment as well as to prevent disease from spreading among all the pigs.

### 2.3. Sample Collection and Analysis

Pigs and feeders were weighed every two weeks to determine average daily gain (ADG), average daily feed intake (ADFI), and feed conversion ratio (FCR). The clinical sign of diarrhea was visually assessed every day by observers blinded to treatments, and a scoring system was a modification of the scoring system previously described by Cappai et al. [14] as follows: 1 = hard feces; 2 = slightly soft feces; 3 = soft, partially formed feces; 4 = loose, semiliquid feces; and 5 = watery. When the average score was less than 3, pigs were identified as having diarrhea, and the diarrhea rate was calculated according to the formula [1]: Diarrhea prevalence = the number of diarrhea pigs × diarrhea days/(the total number of pigs × experiment days) × 100%.

Representative samples of the diets and ingredients were collected and stored at −20 °C until analysis. From day 25 to 27, approximately 50 g of feces were collected daily at random positions from each pen and stored at −20 °C. The three-day collection of feces was pooled by pen and then oven dried at 65 °C for 72 h to determine the apparent total tract digestibility (ATTD) of nutrients according to the equation [1]: ATTD of nutrient = 1 − (Cr_diet_ × Nutrient_feces_)/(Cr_feces_ × Nutrient_diet_). All samples were ground to pass through a 1-mm screen (40 mesh) before analysis. Feed or fecal samples were analyzed for dry matter (DM), crude protein (CP), calcium, and total phosphorus according to the methods of the Association of Official Analytical Chemists (2003) [15]. Gross energy (GE) was determined with an Isoperibol Oxygen Bomb Calorimeter (Parr 6400 Calorimeter, Moline, IL). Chromium content was analyzed using an atomic absorption spectrophotometer (Z-5000 Automatic Absorption Spectrophotometer; Tokyo, Japan). Phosphorus content was analyzed using a UV–vis spectrophotometer (U-1000; Hitachi, Tokyo, Japan).

On d 14 and d 28, 5 mL of blood was drawn from the superior vena cava of selected piglets (*n* = 6/treatment) of average weight using vacutainer tubes (Becton Dickinson Vacutainer Systems, Franklin Lakes, NJ, USA) after an overnight fast. Blood samples were centrifuged at 3000× *g* for 10 min at 4 °C, and serum was then removed and stored at −20 °C until assay. At the end of the experiment, one pig close to the average group body weight was selected from each pen (a total of 18 pigs from three treatments (*n* = 6) for humane slaughter. A small sample of intestinal segments (duodenum, jejunum, and ileum), mucosa (jejunum and ileum), and liver tissue were taken for histology, and then the samples were stored at −80 °C until further analysis.

Intestinal segment samples were embedded in paraffin and cut into 5-μm serial sections, and 5 non-successive sections from each tissue sample were selected and stained with hematoxylin–eosin for histological analysis. Six well-oriented villi (determined as the distance between the crypt openings and the end of the villi) and their associated crypt (measured from the crypt–villous junction to the base of the crypt) per section were selected and measured under a light microscope (CK-40, Olympus, Tokyo, Japan) at 40× magnification and analyzed with an Image Analyzer (Lucia Software. Lucia, ZaDrahou, Czechoslovakia). The average of these measurements was calculated to yield a single value for each pig. These procedures were conducted by an observer unaware of the dietary treatments.

Another sample from each tissue was minced and homogenized (10% w/v) in ice-cold sodium, potassium phosphate buffer (0.01 M, pH 7.4) containing 0.86% NaCl. The homogenate was centrifuged at 3000× *g* for 10 min at 4 °C, and the resultant supernatant was used for analysis. Protein concentration was determined using a Pierce BCA protein assay kit (Nanjing Jiancheng Bioengineering Institute, Nanjing, China) and was expressed as mg/mL. The malondialdehyde (MDA), 8-hydroxy deoxyguanosine (8-OHdG), and carbonyl content were determined using kits from Nanjing Jiancheng Bioengineering Institute (Nanjing, China), and values were expressed in nmol/mg of protein. Determination of superoxide dismutase (SOD), glutathione peroxidase (GSH-Px), catalase (CAT), and total antioxidant capacity (T-AOC) levels were conducted by spectrophotometric methods using a spectrophotometer (Leng Guang SFZ1606017568, Shanghai, China) following the instructions provided by manufacturer (Nanjing Jiancheng Bioengineering Institute, Nanjing, China).

Total RNA was isolated from the liquid nitrogen-pulverized jejunum, ileum, and liver tissue with a TRIZOL Reagent Kit, and then treated with DNase I (Invitrogen, San Diego, CA, USA) according to the manufacturer’s instructions. RNA integrity was tested by 1% agarose gel electrophoresis. The quantity and purity of the RNA samples were measured using a NanoDrop spectrophotometer (Thermo Scientific, Boston, MA) based on the ratio of absorbances at 260 nm and 280 nm. The total RNA was reverse transcribed into cDNA using a PrimeScript RT Reagent Kit (Perfect Real Time; TaKaRa Biotechnology [Dalian] Co. Ltd., Dalian, China). Reverse transcription was performed in a volume of 10 μL including 1 μL cDNA template, 5 μL SYBR Green mix, 3 μL dH_2_O, and 0.5 µmol/L each of forward and reverse primers. Each sample was analyzed in triplicate. The primer information for all of the genes is listed in Table 2. Glyceraldehyde 3-phosphate dehydrogenase (GAPDH) was used as the endogenous control gene in the quantitative RT-PCR experiments. The relative mRNA expression levels were calculated according to the method described by Livak and Schmittgen [16].

### 2.4. Statistical Analysis

All data were analyzed using SAS (Version 9.2, 2008). Differences in the diarrhea prevalence were tested by the chi-square contingency test. All other data were analyzed with an individual pen or pig as the experimental unit using general linear model (GLM) procedures of SAS followed by Tukey’s multiple-range tests. Treatment means were calculated using the least squares means (LSMEANS) statement. The significance level was declared at *p* ≤ 0.05, and trends for significance were declared at *p* < 0.10.

## 3. Results

### 3.1. Growth Performance and Diarrhea Prevalence

During the first two weeks after weaning, there was no difference in ADG, ADFI, and FCR among the treatments (Table 3). Compared with CON, AB and EO improved the ADG of the piglets from days 15 to 28 (*p* < 0.05). Regardless of the period, the diarrhea prevalence of piglets fed AB and EO was lower than that of piglets fed CON (*p* < 0.05). There was no significant difference in growth performance or diarrhea prevalence between AB and EO treatments.

CON = basal diets; AB = basal diets supplemented with 20 mg/kg colistin sulfate and 40 mg/kg zinc bacitracin; EO = basal diets supplemented with 100 mg/kg essential oil blend. ^1^ Values are the means of six observations per treatment, and different superscripts within a row indicate a significant difference (*p* ≤ 0.05).

### 3.2. Faecal Digestibility of Nutrients and Small Intestinal Morphology

Compared with CON, EO increased (*p* < 0.05) the ATTD of gross energy (GE) and CP during the experimental period (Table 4). Consistent with the changes in digestibility of nutrients, villus height in the duodenum and the ratio of villus height to crypt depth in the jejunum for piglets fed AB and EO was greater (*p* < 0.05) than those for piglets fed CON (Table 5).

### 3.3. Serum Oxidant/Antioxidant Status

At day 14, essential oil blends enhanced the SOD and CAT activities and the T-AOC level compared to CON (*p* < 0.05). Consistently, the EO group exhibited lower 8-OHdG content in serum in the first period (*p* < 0.05). However, there was no difference between EO and AB. EO supplementation decreased serum MDA and protein carbonyl content (*p* < 0.05) at day 28 in comparison with CON (Table 6).

### 3.4. Activity and Gene Expression of antioxidant Enzymes

The mucosa in the jejunum of pigs fed EO had greater (*p* < 0.05) T-AOC, SOD, and GSH-Px activities than that of pig-fed CON (Table 7). Essential oil blend tended to increase SOD activity (*p* = 0.09) in the ileum mucosa compared with CON. Pigs fed EO and AB had greater GSH-Px activity in the liver tissue than pigs fed CON (*p* < 0.05). Not only in the jejunum and ileum mucosa, but also in the liver tissue (Figure 1), EO upregulated the SOD1 mRNA expression compared to CON (*p* < 0.05). Meanwhile, the mRNA relative expressions of GPx1 in the ileum mucosa and GPx4 in the liver tissue were higher for pigs fed EO than those fed CON (*p* < 0.05).

## 4. Discussion

Essential oils are natural bioactive compounds with antioxidant properties, which have been approved in the European Union as botanical feed additive [6], and are generally beneficial for animal health and performance [3,12]. In the past years, some progress has been made to demonstrate the positive effect of EO in animal production. However, there some factors, including compositions, dosages, or stages of the animal, that lead to inconsistency and variability. The major components in EO in our study were thymol and cinnamaldehyde. Cinnamaldehyde was shown to exert an antioxidative effect when challenged with cyadox in rabbit erythrocytes [17]. Moreover, thymol has been reported to protect lipid peroxidation in rats [18]. This antioxidant activity could be especially beneficial for piglets during the weaning period, which is stressful on the gut environment and immune response in piglets [9]. In this study, the dietary inclusion of an essential oil blend had no negative effects on feed palatability and intake, which is consistent with previous studies [12,19]. It has been documented that essential oils could improve the growth performance of weaned pigs [20,21]. Our previous studies also proved the higher growth performance and nutrient utilization of weaned pigs supplemented with essential oil blends, which had similar effects to antibiotics [4,11,12]. Also, in the present study, we identified again that the ADG of weaned pigs in the second period and the overall fecal digestibility of nutrients were significantly improved by the essential oil blend and antibiotics, and the pigs supplemented with the essential oil blend had similar performance to those given antibiotics in the present study.

Post-weaning diarrhea is one of the critical factors causing mortality and the retardation of growth in weaned piglets [2]. Diets lacking antibiotics caused an increased occurrence of diarrhea in weaned pigs [4,6], which is consistent with the current study. An essential oil blend significantly reduces diarrhea prevalence and has similarly positive effects compared to antibiotics, which is in agreement with our previous study using thymol and cinnamaldehyde EO [11]. The lower diarrhea prevalence in piglets supplemented with an essential oil blend during the whole experiment period was also consistent with the increased growth performance and ATTD of nutrients in the diet.

The integrity of intestinal morphological structures is crucial for the maintenance of normal intestinal functions. Intestinal villus atrophy or decreases in crypts depth shows that the absorptive ability of the small intestine for nutrients has been decreased [22]. It has been proven that weaning stress results in a sustained impairment of the intestinal barrier and deterioration of the intestinal morphology of pigs [23]. Michiels et al. [24] found that essential oils (carvacrol and thymol) increased the ratio of villus height: crypt depth in the distal small intestine, suggesting an improved gut health of piglets. In our study, EO and AB improved duodenum and jejunum morphology, which was similar to a previous study [4]. Consistent with the studies shown by Zeng et al. (thymol and cinnamaldehyde) [12] and Xu et al. (thymol and carvacrol) [4], our results demonstrated beneficial effects on the intestinal development of weaned pigs resulting in the reduction of diarrhea and the increase of performance and nutrient digestibility. This showed that essential oil blends may alleviate intestinal stress by maintaining or improving small intestinal morphology to enhance the absorption ability and growth performance.

The homeostasis of intracellular reactive oxygen species (ROS) levels is essential in cell proliferation and survival [25]. Young pigs challenged by weaning often suffer from serious oxidative stress [26,27]. This accumulation of ROS may cause damage to macromolecules, including DNA, bio-membrane lipids, and proteins, corresponding with impairment to tissues [28]. Oxidative products of lipids, DNA, and protein injury are generally reflected by MDA, 8-OHdG, and the protein carbonyl content in serum or tissues, respectively. However, these three molecules also act as biomarkers of endogenous peroxidation and radical-induced damages, and generally exhibit different oxidative susceptibilities and durations [26], which may cause the variation of the oxidative products in different tissues in the present results. Not only in the serum, but also in the intestinal mucosa, the essential oil blend can decrease oxidative products, which is similar to previously studies reported by Zeng et al. (thymol and cinnamaldehyde) [12], Zou et al. (oregano) [29], and Wei et al. (thymol and carvacrol) [30].

The antioxidant defense systems consist of non-enzymatic and enzymatic antioxidants for oxidative damage resistance. It is well-known that essential oils have antioxidative effects [3,31] and have been used successfully in animal diets [7,8]. In the present research, the level of T-AOC was increased in serum and jejunum mucosa of pigs supplemented with the essential oil blend, which indicates that the essential oil blend may play an important role in preventing endogenous lipids, DNA, and proteins from peroxidation and oxidation via the non-enzymatic antioxidant defense systems.

Antioxidant enzymes have already been proposed as indicators for monitoring porcine oxidative stress [32]. The SOD, GSH-Px, and CAT stand as the major antioxidant enzymes and are usually regarded as the first-line defense antioxidants [33]. Reactive oxygen species are firstly scavenged by SOD through the reaction 2 H^+^ + 2O_2_^−^ → H_2_O_2_ + O_2_. Hydrogen peroxide is toxic and must be rapidly degraded by subsequent reactions. In mammalian cells, this pathway comprises two major enzyme families: the glutathione peroxidases (GPx) and the catalases (CAT). Both of these two enzyme families detoxify hydrogen peroxide by reducing it to water and oxygen [34]. In the current study, the essential oil blend significantly improved SOD activity in the serum and jejunum mucosa, GSH-Px activity in liver and jejunum mucosa, and CAT activity in serum, which agrees with previous results [12,30]. Consistently, mRNA expression levels of SOD1 in the jejunum and ileum mucosa, and liver tissue, in addition to GPx1 in ileum mucosa and GPx4 in the liver tissue, were significantly increased. Collectively, the decrease in oxidative product levels and the increase in antioxidant defense systems as a result of dietary essential oil inclusion in this study reflected an amelioration of the antioxidant status of weaned pigs. The improvement of antioxidant indices might prevent villi from radical-induced damages, which could explain the better intestinal morphology and nutrient digestibility in pigs administrated EO. In general, the essential oil blend could reduce diarrhea prevalence, improve nutrient absorption, reinforce intestinal morphology, and exert antioxidant properties, which cooperatively lead to the enhanced performance observed in weaned pigs in the present study.

## 5. Conclusions

Overall, dietary essential oil supplementation could decrease diarrhea prevalence and enhance growth performance in weaned pigs through improved antioxidative capacity. Given the similarly positive effects of antibiotics and essential oils on the performance of weaned pigs, the essential oil blend is regarded as a growth promoter in the weaning diet.

## Figures and Tables

**Figure 1 animals-09-00847-f001:**
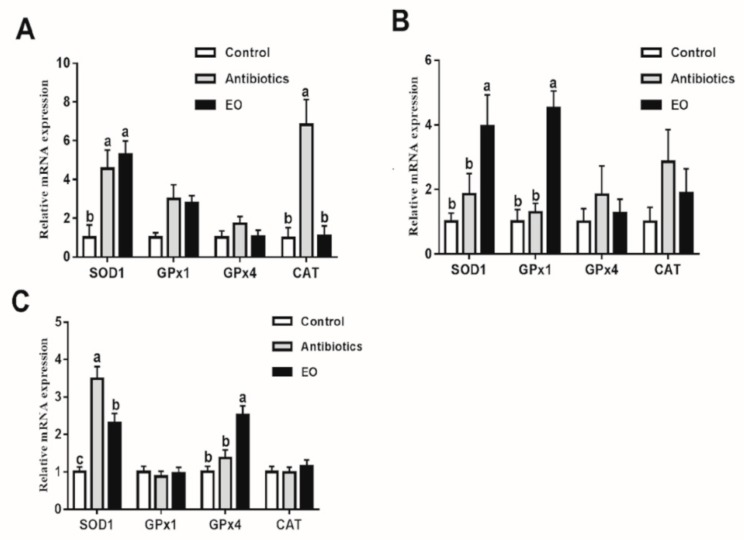
Effects of essential oil supplementation on antioxidant relative mRNA expression of jejunum (**A**) and ileum mucosa (**B**) and liver (**C**) in weaned pigs. Control = basal diets without antibiotics; Antibiotics = basal diets supplemented with 20 mg/kg colistin sulfate and 40 mg/kg zinc bacitracin; EO = basal diets supplemented with 100 mg/kg essential oil blend. The different superscripts within a gene indicate a significant difference (*p* ≤ 0.05). All results are presented as mean ± SEM (*n* = 6).

**Table 1 animals-09-00847-t001:** Composition and nutrient levels of basal diets (g/kg, as-fed).

Item	Days 1–14	Days 15–28
Composition		
Corn	570	643
Soybean meal	60	100
Full fat soybean	50	50
Soy protein concentration	90	60
Fish meal	40	30
Spray-dried plasma protein	30	-
Whey powder	100	60
Soybean oil	29	24.1
Dicalcium phosphate	5.5	5.5
Limestone	10.4	9.5
Sodium chloride	3.0	3.0
L-Lysine-HCl	2.6	4.6
DL-Methionine	0.8	0.8
L-Threonine	0.5	1.5
L-Tryptophan	0.2	0.5
Zinc oxide	3.0	-
Chromic oxide	-	2.5
Premix ^1^	5.0	5.0
Nutrient levels ^2^		
Crude protein	206	188
Calcium	8.1	7.2
Total phosphorus	5.9	5.3
Digestible lysine	13.5	12.4
Digestible methionine	4.0	3.6
Digestible threonine	8.1	7.4
Digestible tryptophan	2.3	2.2
Digestible energy (Mcal/kg)	3.54	3.49

^1^ Premix provided the following per kilogram of feed: vitamin A, 12,000 IU; vitamin D, 2500 IU; vitamin E, 30 IU; vitamin B12, 12 μg; vitamin K, 3 mg; d-pantothenic acid, 15 mg; nicotinic acid, 40 mg; choline chloride, 400 mg; Mn, 40 mg; Zn, 100 mg; Fe, 90 mg; Cu, 8.8 mg; I, 0.35 mg; Se, 0.3 mg. ^2^ Crude protein, calcium, and total phosphorus were analyzed values, and digestible amino acids and energy were calculated using values obtained from the NRC (2012).

**Table 2 animals-09-00847-t002:** The primer sequences for *qRT-PCR* of antioxidant relative mRNA expression of jejunum and ileum mucosa and liver in weaned pigs.

Genes	Orientation	Primer Sequence 5′–3′	Size (bp)	Tm (°C)
GAPDH	Forward	CTGCCGCCTGGAGAAACCT	226	60
	Reverse	GCTGTAGCCAAATTCATTGTCG		
SOD1	Forward	GAGACCTGGGCAATGTGACTG	190	60
	Reverse	GCCAAACGACTTCCAGCAT		
GPX1	Forward	CAGGCGGCGGGTTCG	129	60
	Reverse	TGAGGGCAGTGGCATCGT		
GPX4	Forward	GAAGGACCTGCCGTGCTACC	159	60
	Reverse	CCTTCCTCTGGCGGGGTT		
CAT	Forward	CCCGCATTCAGGCTCTTC	116	60
	Reverse	CCCTCACAGGTTAGCTTTCTCC		

Tm = annealing temperature.

**Table 3 animals-09-00847-t003:** Effects of essential oil supplementation on the growth performance and diarrhea prevalence of weaned piglets ^1^.

Item	CON	AB	EO	SEM	*p*-Value
Days 1–14					
Average daily gain, g	283	305	301	19.7	0.71
Average daily feed intake, g	453	465	447	33.9	0.93
Feed conversion ratio	1.6	1.5	1.5	0.05	0.32
Diarrhea prevalence, %	17.4 ^a^	11.2 ^b^	12.6 ^b^	0.78	<0.01
Days 15–28					
Average daily gain, g	405 ^b^	475 ^a^	462 ^a^	17.8	0.03
Average daily feed intake, g	747	840	802	35.8	0.21
Feed conversion ratio	1.8	1.8	1.7	0.04	0.25
Diarrhea prevalence, %	9.3 ^a^	4.8 ^b^	5.7 ^b^	0.69	<0.01
Days 1–28					
Average daily gain, g	344	390	381	18.3	0.20
Average daily feed intake, g	599	653	624	33.5	0.54
Feed conversion ratio	1.7	1.7	1.6	0.04	0.15
Diarrhea prevalence, %	13.3 ^a^	8.0 ^b^	9.2 ^b^	0.45	<0.01

CON = basal diets; AB = basal diets supplemented with 20 mg/kg colistin sulfate and 40 mg/kg zinc bacitracin; EO = basal diets supplemented with 100 mg/kg essential oil blend. ^1^ Values are the means of six observations per treatment, and different superscripts within a row indicate a significant difference (*p* ≤ 0.05).

**Table 4 animals-09-00847-t004:** Effects of essential oil supplementation on apparent total tract digestibility of nutrients in weaned piglets on day 28 in the experiment ^1^.

Item	CON	AB	EO	SEM	*p*-Value
Gross energy	0.79 ^b^	0.80 ^ab^	0.81 ^a^	0.001	0.04
Dry matter	0.81	0.81	0.81	0.005	0.65
Crude protein	0.75 ^b^	0.78 ^a^	0.78 ^a^	0.008	0.04

CON = basal diets; AB = basal diets supplemented with 20 mg/kg colistin sulfate and 40 mg/kg zinc bacitracin; EO = basal diets supplemented with 100 mg/kg essential oil blend.^1^ Values are means of six observations per treatment, and different superscripts within a row indicate a significant difference (*p* ≤ 0.05).

**Table 5 animals-09-00847-t005:** Effects of essential oil supplementation on small intestine morphology of weaned piglets ^1^.

Item	CON	AB	EO	SEM	*p*-Value
Duodenum					
Villus height, μm	34 ^b^	414 ^a^	407 ^a^	16.4	0.02
Crypt depth, μm	243	240	251	18.7	0.92
Villus height: Crypt depth	1.43	1.74	1.63	0.09	0.08
Jejunum					
Villus height, μm	376	377	410	25.3	0.57
Crypt depth, μm	249	214	228	18.8	0.43
Villus height: Crypt depth	1.51 ^b^	1.76 ^a^	1.82 ^a^	0.05	<0.01
Ileum					
Villus height, μm	271	303	280	21.0	0.56
Crypt depth, μm	188	197	189	17.8	0.92
Villus height: Crypt depth	1.48	1.54	1.49	0.09	0.87

CON = basal diets; AB = basal diets supplemented with 20 mg/kg colistin sulfate and 40 mg/kg zinc bacitracin; EO = basal diets supplemented with 100 mg/kg essential oil blend. ^1^ Values are means of six observations per treatment, and different superscripts within a row indicate a significant difference (*p* ≤ 0.05).

**Table 6 animals-09-00847-t006:** Effects of essential oil supplementation on antioxidant enzyme activity and oxidant products of serum in weaned piglets ^1^.

Item	CON	AB	EO	SEM	*p*-Value
Day 14					
Superoxide dismutase (SOD), U/mL	71.1 ^b^	75.9 ^b^	82.0 ^a^	1.64	<0.01
Glutathione peroxidase (GPX), U/mL	703	757	751	22.4	0.20
Catalase (CAT), U/mL	5.31 ^b^	6.08 ^ab^	6.65 ^a^	0.33	0.04
Total antioxidant capacity (T-AOC), U/mL	8.52 ^b^	9.84 ^b^	19.64 ^a^	2.73	0.02
Malondialdehyde (MDA), nmol/mg	11.57	7.31	8.02	1.43	0.11
8-hydroxy deoxyguanosine (8-OHdG), ng/mg	30.5 ^a^	25.2 ^ab^	19.3 ^b^	2.48	0.02
Protein carbonyl, nmol/mg	1.69	1.22	1.24	0.35	0.58
Day 28					
Superoxide dismutase (SOD), U/mL	81.1	86.6	83.9	2.98	0.44
Glutathione peroxidase (GPX), U/mL	709	750	724	29.6	0.63
Catalase (CAT), U/mL	5.39	5.83	5.20	0.43	0.58
Total antioxidant capacity (T-AOC), U/mL	33.5	46.0	37.6	3.63	0.07
Malondialdehyde (MDA), nmol/mg	8.82 ^a^	4.89 ^b^	5.87 ^b^	0.73	<0.01
8-hydroxy deoxyguanosine (8-OHdG), ng/mg	23.0	24.4	21.4	2.78	0.74
Protein carbonyl, nmol/mg	1.23 ^a^	0.96 ^ab^	0.46 ^b^	0.20	0.05

CON = basal diets; AB = basal diets supplemented with 20 mg/kg colistin sulfate and 40 mg/kg zinc bacitracin; EO = basal diets supplemented with 100 mg/kg essential oil blend. ^1^ Values are the means of six observations per treatment, and different superscripts within a row indicate a significant difference (*p* ≤ 0.05).

**Table 7 animals-09-00847-t007:** Effects of essential oil supplementation on antioxidant enzyme activity and oxidant products of intestine mucosa and liver in weaned piglets ^1^.

Item	CON	AB	EO	SEM	*p*-Value
Jejunum mucosa					
Superoxide dismutase (SOD), U/mL	642 ^b^	820 ^ab^	964 ^a^	57.7	0.01
Glutathione peroxidase (GPX), U/mL	1024 ^b^	1260 ^a^	1381 ^a^	65.2	0.01
Catalase (CAT), U/mL	86.0	83.9	82.1	13.6	0.98
Total antioxidant capacity (T-AOC), U/mL	115 ^b^	124 ^b^	197 ^a^	17.2	0.02
Malondialdehyde (MDA), nmol/mg	4.48	5.70	6.20	1.23	0.61
8-hydroxy deoxyguanosine (8-OHdG), ng/mg	47.7	41.4	49.4	5.63	0.58
Protein carbonyl, nmol/mg	50.3	39.9	48.9	2.96	0.07
Ileum mucosa					
Superoxide dismutase (SOD), U/mL	737	784	927	55.7	0.09
Glutathione peroxidase (GPX), U/mL	1283	1311	1440	89.1	0.45
Catalase (CAT), U/mL	48.3	28.4	36.9	9.51	0.37
Total antioxidant capacity (T-AOC), U/mL	133	152	172	16.0	0.27
Malondialdehyde, nmol/mg	11.18	5.84	9.69	1.54	0.09
8-hydroxy deoxyguanosine (8-OHdG), ng/mg	91.5	97.3	98.4	9.37	0.86
Protein carbonyl, nmol/mg	66.3	70.1	69.4	5.37	0.87
Liver					
Superoxide dismutase (SOD), U/mL	102	118	121	9.17	0.31
Glutathione peroxidase (GPX), U/mL	552^b^	648^a^	637^a^	24.1	0.05
Catalase (CAT), U/mL	18.8	24.3	22.8	3.99	0.62
Total antioxidant capacity (T-AOC), U/mL	76.8	87.9	91.0	9.97	0.59
Malondialdehyde (MDA), nmol/mg	8.18	5.59	7.15	0.91	0.18
8-hydroxy deoxyguanosine (8-OHdG), ng/mg	6.93	6.00	7.60	0.92	0.49
Protein carbonyl, nmol/mg	8.80	5.61	6.94	1.16	0.21

CON = basal diets; AB = basal diets supplemented with 20 mg/kg colistin sulfate and 40 mg/kg zinc bacitracin; EO = basal diets supplemented with 100 mg/kg essential oil blend.^1^ Values are the means of six observations per treatment, and different superscripts within a row indicate a significant difference (*p* ≤ 0.05).

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
