# Peer review of "Essential Oil Blend Could Decrease Diarrhea Prevalence by Improving Antioxidative Capability for Weaned Pigs"

_animals, 2019, doi:10.3390/ani9100847_

Round 1
Reviewer 1 Report
Dear Authors,
I evaluated the manuscript and I believe it is well presented and written. In the attached file you will find some suggestions to further improve it.
However I have a serious concern with respect to this paper: as it stands, it looks more as a commercial trial aimed at assessing the effectiveness of an additive than as a research paper aimed at exploring possible alternatives to antibiotic use.
For example, the characteristics of the product used are never analyzed thoroughly or described, neither the expected mechanism of action is described. When Authors compare with previous studies (from their or from other research groups) it is totally unclear if they refer to somehow similar essential oils formulations or to completely different ones. As there might be a huge variability in botanical composition, synergies and effectiveness, unless better specified all these comparisons made in the manuscript appear, to the least, very weak. For example, in some studies the Authors refer to the use of "carvacrol and thymol" whereas this study is appartently on "cinnamaldehyde and thymol", but the difference is never highighted in the Discussion section.
I suggest Authors to throughly rethink their work in order to give to it a more to-the point, factual shape. Author seem to forget that they are testing a single commercial product an not "dietary essential oil supplementation", as stated for example in the Conclusions section.
lastly, authors should also comment and explain on why they chose such a low number of experimental units.

Author Response
Point 1: I evaluated the manuscript and I believe it is well presented and written. In the attached file you will find some suggestions to further improve it.
However I have a serious concern with respect to this paper: as it stands, it looks more as a commercial trial aimed at assessing the effectiveness of an additive than as a research paper aimed at exploring possible alternatives to antibiotic use.
Response 1: Thanks for your comments. In this study we used antibiotics as the positive control to assess the effect of EO on animal performance. Our data indicated that EO supplementation increased growth performance, decreased diarrhea score and improved oxidative status in weaned piglets. Per suggestions, we have revised the manuscript accordingly.
Point 2: For example, the characteristics of the product used are never analyzed thoroughly or described, neither the expected mechanism of action is described. When Authors compare with previous studies (from their or from other research groups) it is totally unclear if they refer to somehow similar essential oils formulations or to completely different ones. As there might be a huge variability in botanical composition, synergies and effectiveness, unless better specified all these comparisons made in the manuscript appear, to the least, very weak. For example, in some studies the Authors refer to the use of "carvacrol and thymol" whereas this study is appartently on "cinnamaldehyde and thymol", but the difference is never highighted in the Discussion section.
I suggest Authors to throughly rethink their work in order to give to it a more to-the point, factual shape. Author seem to forget that they are testing a single commercial product an not "dietary essential oil supplementation", as stated for example in the Conclusions section.
Response 2: Thanks for these suggestions. We agree that the botanical composition or concentration of EO causes a huge variability. So, we added the information of those major components in the manuscripts. In addition, we have revised the manuscript accordingly in Discussion and Conclusion section.
Point 3: lastly, authors should also comment and explain on why they chose such a low number of experimental units.
Response 3: Thanks for your comments. we have tested the normality of data distribution. Though a low number of animals has been used in this study. Our lab did conduct EO treatments several times to make sure the results are replicable. We pooled the data together and tested the normality of data distribution using the “Proc univariate” of the SAS program, which confirmed the normal distribution of data and equal variances. The p values of Kolmogorov-Simirnov, Cramer-von Mises and Anderson-Darling were all greater than 0.05.
Reviewer 2 Report
Dear authors
I read with interest your piece of work and I found one blank point, for which I ask to provide details. I was wondering whether you purchased the basal diet or formulated and produces by yourselves. Why do you choose to use only corn? Is there a particular reason? Which from of the diet did you use? How did you introduce re EO blend to prevent peroxidation? Please, add detailed information. How was the dose decided? did you carry out titration trals? Was the EO diet palatable to piglets? Were they used to the taste prior to start the experimental feeding? How long were they adapted? Was the use of AB diet used only for experimental purpose or also for treatment of animals? Thanks.
Author Response
Point 1: I read with interest your piece of work and I found one blank point, for which I ask to provide details. I was wondering whether you purchased the basal diet or formulated and produces by yourselves. Why do you choose to use only corn? Is there a particular reason? Which from of the diet did you use? How did you introduce re EO blend to prevent peroxidation? Please, add detailed information. How was the dose decided? did you carry out titration trals? Was the EO diet palatable to piglets? Were they used to the taste prior to start the experimental feeding? How long were they adapted? Was the use of AB diet used only for experimental purpose or also for treatment of animals? Thanks.
Response 1: Thanks for your comments. In the current study we formulated our diet which met the nutrient requirements of swine (NRC, 2012). Corn-based diets are widely used in China. So, in this study we also used corn-based diet.
Because the EO blend used in our study was encapsulated in a maltodextrin matrix which prevent the oxidation of those bioactive compounds. This information has been added into the manuscript.
Our lab did several studies in the past years. So, the dose of EO was determined by our preliminary study and the recommendation provided by producer.
In this study we did not have an adaptation period. However, our data indicated EO supplementation did not affect the feed intake. It is probably because those ingredients were encapsulated in the maltodextrin matrix. So, that would not affect the taste too much.
Antibiotics have been used as growth promoters in animal production. However, those antibiotics applied in our study are banned for swine industry. Weaned piglets are very susceptible to pathogens. So, here we used AB as a positive control in order to evaluate the effect of EO on growth performance.
Round 2
Reviewer 1 Report
Dear Authors,
thanks for revising the manuscript according to my suggestions.
Author Response
Dear Authors,
thanks for revising the manuscript according to my suggestions.
Response: Thank you for all your comments and suggestions.
Reviewer 2 Report
Dear Authors,
I reviewed the revised version of your manuscript and I found great merit in your research. However, as stated in the previous round of review you speak of diarrhoea whereas this is incorrect. What you do is to judge the appearance of faeces for which you can refer to Cappai et al. 2013, Journal of Animal Physiology ad Animal Nutrition. Then, clinically speaking, after you state that faeces are 'watery' you can point to the different types of diarrhoea (hemorrhagic? Mucous-Croupal?).
What you describe CANNOT be reported as diarrhoea score because it is prevalence over time:apparently hit/apparently healthy x 100.
What you reported is wrong.
I strongly recommend amends!
Author Response
Point 1: I reviewed the revised version of your manuscript and I found great merit in your research. However, as stated in the previous round of review you speak of diarrhoea whereas this is incorrect. What you do is to judge the appearance of faeces for which you can refer to Cappai et al. 2013, Journal of Animal Physiology and Animal Nutrition. Then, clinically speaking, after you state that faeces are 'watery' you can point to the different types of diarrhoea (hemorrhagic? Mucous-Croupal?).
Response 1: Thank you for your comments. The scoring system has been amended according to Cappai et al. 2013. We have also revised the manuscript accordingly.
Point 2: What you describe CANNOT be reported as diarrhoea score because it is prevalence over time:apparently hit/apparently healthy x 100.
What you reported is wrong.
Response 1: Thanks for your comments. ‘Diarrhoea score’ has been changed to ‘diarrhoea prevalence’